# Long-Term Impact of Deep Brain Stimulation in Parkinson’s Disease: Does It Affect Rehabilitation Outcomes?

**DOI:** 10.3390/medicina60060927

**Published:** 2024-06-01

**Authors:** Margherita Canesi, Lorenzo Lippi, Simone Rivaroli, Daniele Vavassori, Marta Trenti, Francesco Sartorio, Nicoletta Meucci, Alessandro de Sire, Chiara Siri, Marco Invernizzi

**Affiliations:** 1U.O.C of Neurorehabilitation, Parkinson’s Disease and Movement Disorders Center, Moriggia Pelascini Hospital, Gravedona ed Unit, 22015 Como, Italy; margheritag2007@yahoo.it (M.C.); daniele.vavassori.fisio@gmail.com (D.V.); nicoletta.meucci@ospedaledigravedona.it (N.M.); chiarasiri.park@gmail.com (C.S.); 2Department of Scientific Research, Campus LUdeS Lugano (CH), Off-Campus Semmelweis University of Budapest, 1085 Budapest, Hungary; marta.trenti@gmail.com (M.T.); francesco.sartorio@uniludes.ch (F.S.); 3Department of Health Sciences, University of Eastern Piedmont “A. Avogadro”, 28100 Novara, Italy; srivaroli@gmail.com (S.R.); marco.invernizzi@med.uniupo.it (M.I.); 4Department of Medical and Surgical Sciences, University of Catanzaro “Magna Graecia”, 88100 Catanzaro, Italy; alessandro.desire@unicz.it; 5Research Center on Musculoskeletal Health, MusculoSkeletalHealth@UMG, University of Catanzaro “Magna Graecia”, 88100 Catanzaro, Italy

**Keywords:** Parkinson Disease, Deep Brain Stimulation, rehabilitation, motor skills disorders

## Abstract

*Background and Objectives*: Although the growing literature is now focusing on the long-term effects of Deep Brain Stimulation (DBS) in Parkinson’s disease (PD), there is still a large gap of knowledge about its long-term implications in rehabilitation. Therefore, this study aimed at investigating the effects of rehabilitation in PD patients years after DBS implantation. *Materials and Methods:* This retrospective case–control study analyzed records from Moriggia-Pelascini Hospital, Italy from September 2022 to January 2024. Data of PD patients (*n* = 47) with (DBS group, *n* = 22) and without (control group, *n* = 25) DBS were considered. All study participants underwent a daily rehabilitation program lasting four weeks, including warm-up, aerobic exercises, strength training, postural exercises, and proprioceptive activities. The outcomes assessed were the Unified Parkinson’s Disease Rating Scale (UPDRS), Berg Balance Scale (BBS), Timed Up and Go (TUG), 6 Min Walk Test (6MWT), and Self-Assessment Parkinson Disease Scale (SPDDS). *Results:* DBS group showed significant improvements in terms of all outcome measures after the rehabilitation intervention (UPDRS III: −7.0 (−11.5 to −1.0); *p* = 0.001; UPDRS I II IV: −12.0 (−19.0 to −4.5); *p* = 0.001; BBS: 7.0 (3.8 to 10.3); *p* < 0.001; TUG (s): −2.8 (−5.7 to −1.1); *p* < 0.001; SPDDS: −8 (−13.0 to −4.0); *p* < 0.001; 6MWT (m): 81 (37.3 to 132.3); *p* < 0.001). No differences were reported in the between-group analysis (p: NS). *Conclusions*: This study emphasizes positive rehabilitation effects on PD patients irrespective of DBS status. Further research is essential to elucidate long-term effects of DBS on rehabilitation outcomes of PD patients.

## 1. Introduction

Parkinson’s disease (PD) is a neurodegenerative disorder characterized by motor and non-motor symptoms [1], significantly impacting the quality of life of millions of people worldwide [2]. Among the motor symptoms of PD, tremors, bradykinesia, rigidity, postural instability, and freezing are commonly observed [3]. These symptoms result from the progressive degeneration of dopaminergic neurons in the substantia nigra and significantly impact the physical function of patients with PD [4]. Moreover, non-motor symptoms of PD might include cognitive impairment, psychiatric symptoms such as depression and anxiety, autonomic dysfunction affecting blood pressure and digestion, and sleep disorders [5]. These symptoms often arise from more widespread pathology and can significantly impact daily functioning and quality of life for individuals with PD [6].

Unfortunately, PD prevalence continues to rise due to the aging of the population, becoming a significant global health concern [7,8]. In particular, a recent study [7] reported a global increase in the age-standardized rates of incidence, prevalence, and years lived with disability with an estimated annual percentage change of 0.61 (95% confidence interval [CI]: 0.58–0.65), 0.52 (95% CI: 0.43–0.61), and 0.53 (95% CI: 0.44–0.62) respectively.

Taken together, effective strategies reducing functional implications of PD are necessary, with the growing literature supporting the effects of innovative approaches to improve patient outcomes [9,10].

Beyond pharmacological approaches, rehabilitation plays a pivotal role in managing the complex disability characterizing these patients [11,12,13]. Tailored rehabilitation programs, including physical, occupational, and speech therapies, aim not only to address motor symptoms but also to enhance overall functional capacity and mitigate the impact of non-motor symptoms [14]. Other non-pharmacological approaches might be integrated into the therapeutic management of PD, including Deep Brain Stimulation (DBS), which has emerged as a transformative intervention for managing motor symptoms in advanced PD [15]. DBS involves the surgical implantation of electrodes into specific regions of the central nervous system, including the subthalamic nucleus (STN) or globus pallidus interna (GPi) [16]. The electrical stimulation of these brain regions modulates abnormal neuronal activity providing positive effects in terms of motor symptoms not responsive to conventional medications [16]. The electrical stimulation of specific brain regions provides therapeutic relief, yet the broader implications of DBS on functional outcomes remain a subject of ongoing exploration [15,17].

Interestingly, the recent study by Hitti et al. [18] underlined a long-term improvement of motor symptom control and ability to perform activities of daily living (ADLs) in patients with PD undergoing DBS, although it does not halt disease progression. On the other hand, the recent meta-analysis by Bucur and Papagno [15] reported positive long-term effects of DBS also in terms of anxiety and depression, while negative consequences in cognitive function were reported, including a decline in long-term memory and phonemic verbal fluency.

While growing research is now focusing on DBS’s long-term outcomes [19,20], recognizing the complementary benefits of rehabilitation post-DBS is crucial [21]. Understanding the potential synergisms between DBS and rehabilitation could offer novel insights into optimizing the overall functional recovery in PD patients. In line with this consideration, in 2022 Sato et al. [21] assessed the role of a postoperative rehabilitation combined with DBS. The authors reported a short-term improvement in terms of gait step length and trunk function when compared to the preoperative medicated state, with intriguing implications in postoperative postural stability [21].

However, there is still a large gap in knowledge of the long-term effects of DBS patients receiving a rehabilitation treatment. In this context, the reported impaired cognitive function in patients with DBS might have negative implications for rehabilitation outcomes [15]. In contrast, improved motor function with DBS might provide decreased efficacy of a tailored rehabilitation intervention in patients with PD [22].

However, to the best of our knowledge, no previous study assessed the role of a specific rehabilitation intervention in PD patients after years from DBS implantation.

Thus, this retrospective case–control study aims to assess the effects of a specific rehabilitation intervention in patients with PD after years from DBS, characterizing its role in terms of physical functioning, physical performance, and independence in ADLs. These findings might guide clinicians in an evidence-based prescription of a specific rehabilitation intervention in patients with PD after years from DBS.

## 2. Materials and Methods

### 2.1. Study Design and Ethics

This retrospective case–control study conforms to the “Strengthening the Reporting of Observational Studies in Epidemiology” (STROBE) Guidelines [23]. In this study, we assessed patients with PD with and without Deep Brain Stimulation (DBS) who underwent a specific rehabilitation program. The study design involved the retrospective analysis of medical records from Moriggia-Pelascini Hospital of Gravedona (Italy) between September 2022 and January 2024.

The study protocol followed the ethical principles outlined in the Declaration of Helsinki [24]. Privacy protection was guaranteed by study researchers throughout the study. Personal data were anonymized, and access to sensitive patient information was restricted to authorized personnel. All participants included in the study had previously signed informed consent forms, explicitly granting permission for the use of their medical records for research purposes.

### 2.2. Study Participants

We included patients meeting the following inclusion criteria: (i) patients diagnosed with PD based on MDS clinical diagnostic criteria for PD [25]; (ii) Stage 2 or 3 according to the Hoehn and Yahr scale [26]; (iii) patients who participated in a specific rehabilitation program designed for PD; (iv) availability of complete medical records; (v) signed informed consent.

Exclusion criteria were the following: (i) surgical procedures in the 6 months before rehabilitation; (ii) neurological conditions other than PD; (iii) cognitive impairment assessed by Mini-Mental State Examination (MMSE) score less than 24 [27]; (iv) uncontrolled arrhythmias, unstable angina, deep vein thrombosis, or other contraindications to physical rehabilitation; (v) individuals unable to perform physical testing.

In the study, eligible patients were systematically divided into two different groups: the DBS Group and the no-DBS Group. This categorization was performed by the presence or absence of DBS. Both groups underwent the same inpatient rehabilitation program specifically designed for patients with PD.

### 2.3. Rehabilitation Protocol

After baseline assessment, all the patients underwent the same rehabilitation intervention divided into two sessions per day, each lasting approximately 60 min conducted five days a week, extending over a duration of four weeks. As a result, all the study participants were engaged in 24 pairs of semi-standardized rehabilitative sessions, focused on different domains characterizing the multicomponent disability of patients with PD.

The morning session included a warm-up phase lasting 10 min of passive and active mobilization exercises for both upper and lower limbs. Subsequently, 15 min of aerobic exercises were performed aiming at enhancing the task-oriented patients’ endurance in walking function. Aerobic activities included walking and cycling with a focus on maintaining exercise intensity between 50% and 80% of the maximal heart rate, tailored to each patient’s individual tolerance. The exercise protocol was designed to facilitate gradual progressions throughout the intervention days. Subsequently, 15 min of active mobilization exercises and strengthening exercises were performed to increase muscle strength and improve physical performance. More in detail, light weightlifting, TheraBand^®^ (Hygenic Corp. Performance Health, Akron, OH, USA) resistance exercise, and free weights exercise were used, focusing on all primary muscle groups with a resistance set at 60–75% of the estimated one-repetition maximum (1RM). Moreover, 10 min of postural exercises targeting camptocormic posture, preventing body flexion, and addressing instability were performed, combined with proprioceptive exercises with dual-task components in order to work on balance function in both static and dynamic conditions. Lastly, a 10 min cooling down phase was performed with passive and active mobilization exercises.

The afternoon session included a short warm-up of 10 min followed by active exercise treatments through treadmill, MOTOmed (Chinesport SpA, Udine, Italy), and stabilometric platforms. More in detail, it included a 15 min session on the treadmill, paying attention to the walking movement pattern quality. In addition, 15 min of aerobic exercises were performed on the MOTOmed. Aerobic exercise intensity was set between 50% and 80% of the maximal heart rate. The afternoon session was concluded with 15 min of proprioceptive exercises, utilizing the stabilometric platform in both static and dynamic conditions. Lastly, a 5 min cooling down phase was performed with passive and active mobilization exercises. All rehabilitation sessions were supervised by an expert physiotherapy with years of expertise in PD rehabilitation. The therapist/patient rate was one-to-one. Adherence to the rehabilitation program was recorded by tracking session participation. A minimum compliance rate of 80% was required for inclusion in the data analysis.

In addition, the comprehensive rehabilitation approach was combined with conventional occupational therapy sessions to enhance patients’ independence in ADL and quality of life. Each session lasted one hour per day, five days a week, and included exercises targeting fine and gross motor skills, coordination, and balance. The occupational therapist focused on activities such as dressing, grooming, and meal preparation, tailoring interventions to the specific needs and abilities of the individual. Cognitive strategies and adaptive techniques have been incorporated to improve overall functional capacity. On the other hand, all the patients underwent conventional speech–language sessions for Parkinson’s disease, including a comprehensive approach to address communication challenges. These sessions included exercises targeting voice modulation, articulation, and speech clarity. In addition, cognitive-communication strategies and exercises to improve language comprehension and expression were included.

### 2.4. Outcomes

An expert physician specialized in physical and rehabilitation medicine assessed clinical history, sociodemographic, anthropometric information, and outcome scales of each study participant. In addition, an expert physiotherapist evaluated all performance tests. Clinical history, sociodemographic, and anthropometric information were assessed at T0, while primary and secondary outcomes measures were evaluated at each timepoint.

The outcomes were:-*Movement Disorder Society—Unified Parkinson’s Disease Rating Scale (MDS-UPDRS)* [28]: This is a comprehensive assessment tool used to assess the severity and progression of PD. Originally developed in 1987 and later revised, the MDS-UPDRS consists of four parts (Part I: Mentation, Behavior, and Mood; Part II: Activities of Daily Living (ADL); Part III: Motor Examination; Part IV: Complications of Therapy). Overall, the MDS-UPDRS provides a standardized and systematic way to assess both motor and non-motor aspects of PD [28]. In order to characterize motor symptoms, we considered MDS-UPDRS III, while for a comprehensive analysis of non-motor symptoms, we considered MDS-UPDRS I II and IV.-*Berg Balance Scale (BBS)* [29]: This is a clinical assessment tool to assess the static and dynamic balance abilities. It includes 14 functional tasks, each graded on a five-point scale based on the individual’s performance. The tasks assess various aspects of balance, ranging from sitting and standing to more dynamic activities like reaching and turning. The BBS is particularly relevant in assessing balance impairments in neurological conditions such as PD. Changes in BBS scores over time can help track the progression of balance impairments or assess the effectiveness of interventions [29].-*Self-Assessment Parkinson Disease Scale (SPDDS)* [30]: This is a clinical outcome measure for a comprehensive assessment of ADL. The patient rates their ability to perform ADL on a five-point scale, ranging from “able to do alone without difficulty” to “unable to do at all”. The questionnaire includes a total of 25 items for a comprehensive assessment of patients with PD [30].-*Timed Up and Go (TUG) test* [31]: This is a functional mobility and dynamic balance test used in clinical settings to assess mobility and risk of falls. The TUG test begins with the patient seated in a standard armchair. On the command “Go,” the patient stands up from the chair and walks a distance of 3 m, turns around, and walks back to the chair to sit down. The total time taken to complete the task is recorded and provides valuable information about a person’s mobility, balance, and gait. The TUG test is considered a quick and reliable measure to assess changes in performance over time may guide interventions and treatment plans for individuals with PD [31].-*6-Min Walk Test (6MWT)* [32]: This is a clinical assessment tool to evaluate functional exercise capacity and endurance in patients with different conditions. The test measures the distance a person can walk in six minutes and is commonly employed to assess cardiovascular and respiratory function, as well as overall mobility and endurance. The patient was instructed to walk as far as possible in a straight path for six minutes. The total distance covered in six minutes was recorded. Heart rate and oxygen saturation were monitored before, during, and after the test [32].-*Montreal Cognitive Assessment (MoCA)* [33]: a widely used cognitive screening tool designed to assess different cognitive domains, including memory, attention, language, visuospatial skills, and executive function. The outcome measure of the MoCA is a numerical score, with a maximum score of 30 points. A higher score indicates better cognitive functioning, while a lower score may suggest potential cognitive impairment [33].

All the outcome measures were assessed at the baseline (T0) and 24 h after the end of rehabilitation treatment (T1). All the complications that occurred during hospitalization were registered in both study groups to assess the safety of the study intervention.

### 2.5. Statistical Analysis

The statistical analyses were performed using GraphPad Prism 7.0 (GraphPad Software, Inc., San Diego, CA, USA). Descriptive statistics were used to summarize demographic and clinical characteristics of the study groups. Categorical variables were presented using numbers and ratios, while continuous variables were expressed as median with interquartile ranges (IQR). The normality distribution of the variables was assessed using the Shapiro–Wilk statistic. Intragroup differences at different time points were evaluated using Wilcoxon’s signed-rank test. Comparative analysis between DBS Group and non–DBS Group was performed using the Mann–Whitney U test. Statistical significance was set at a *p*-value below 0.05. Spearman’s rank correlation coefficient (r) with 95% confidence intervals (CI) was employed to evaluate the correlation among different clinical variables.

## 3. Results

Out of the 59 patients screened for eligibility, 12 did not meet the eligibility criteria (3 had missing data, 5 did not sign the informed consent or declined to participate, 1 patient had cardiovascular disorders, and 3 had gastrointestinal complications that affected adherence to the rehabilitation program). Therefore, 47 patients (33 males and 14 females), were included in the data analysis and subsequently divided into two groups. Further details of the screening selection process are shown in Figure 1.

Eligible patients were systematically divided into two groups: the DBS group and the no-DBS group. A total of 22 patients (17 males and 5 females) were included in the DBS group (median age: 63.5 years; Interquartile Range (IQR) 57.0 to 70.5 years), while 25 patients (16 males and 9 females) were included in no-DBS group (median age: 69.0 years, IQR 63.0 to 74.0). In the DBS group, the median age at the DBS implant was 56.0 years (IQR 50.8 to 61.5), the median years between PD diagnosis and DBS implant was 10.0 years (IQR 8.8 to 14.0), and the median time between DBS implant and the rehabilitation intervention was 6.0 years (IQR 5.0 to 10.8). No significant differences were shown in the baseline characteristics between the DBS group and no-DBS group in terms of sex, age, ethnicity, disease duration, levodopa dose, Hoehn and Yahr Stage, and MOCA score. The characteristics of the study participants are described in Table 1.

After the rehabilitation treatment, significant improvements were reported in DBS group in terms of physical functioning and physical performance assessed by MDS-UPDRS III scores, TUG, and 6MWT. On the other hand, significant improvements were reported also in terms of balance function and independence in ADL, as shown by the BBS, MDS-UPDRS I II IV, and SPPDS scores. In accordance, the no-DBS group showed similar improvement after rehabilitation, as highlighted by significant changes over time in MDS-UPDRS, BBS, TUG, PPDS, and 6MWT scores. Further details about the within-group differences were reported in Table 2. Moreover, there were no statistical differences between the DBS group and the control group in terms of improvements for all the outcome measures (see Table 2 for further details).

In DBS group there was a significant negative correlation between time from DBS intervention and 6MWT improvements (r = −0.516, *p* = 0.014), while a statistically significant negative correlation was reported between the age of PD patients and the 6MWT improvements (r = −0.427, *p* = 0.047). Moreover, there was a statistically significant positive correlation (r = 0.434, *p* = 0.044) between the duration from diagnosis and DBS, and TUG. The results of the correlation analysis are shown in Figure 2.

Lastly, a partial correlation analysis was performed according to age and included in Appendix A.

## 4. Discussion

In recent years, attention on the long-term effects of DBS in patients with PD has been growing. While its effects on motor and non-motor functions are still controversial [15,17,18], there are no data about the effects of rehabilitation in PD patients years after DBS implantation. Therefore, this retrospective case–control study explores the effects of specific rehabilitation intervention in PD patients post-DBS implantation, focusing on physical functioning, physical performance, and independence in ADLs.

Interestingly, our findings underlined significant improvements in key outcome measures, including MDS-UPDRS. The improvements in MDS-UPDRS scores in PD patients with DBS reflect a holistic improvement of motor and non-motor functioning, supporting a positive impact on the global functional status of individuals with PD. More in detail, significant effects were reported in terms of MDS-UPDRS III (Median (IQR): −7.0 (−11.5 to −1.0); *p*: 0.001) and MDS-UPDRS I II IV (Median (IQR): −12.0 (−19.0 to −4.5); *p* = 0.001). In accordance, positive effects were reported also in terms of balance functioning, as underlined by the BBS (7.0 (3.8 to 10.3); *p* < 0.0001) and TUG test (Median (IQR): −2.8 (−5.7 to −1.1); *p* < 0.0001). Both the DBS and control groups exhibited significant improvements in BBS scores, indicative of enhanced static and dynamic balance abilities. The concurrent reduction in TUG test suggested relevant implications for functional mobility. Moreover, different reports underlined the strict relation between TUG results and fall risk [34,35]. In addition, PD patients might be frequently characterized by increased risk of fracture due to the high prevalence of osteoporosis and sarcopenia [36,37,38]. Thus, a multitarget intervention focusing not only on risk of fall prevention and balance function but targeting musculoskeletal health is necessary to reduce the fragility fracture risk and improve long-term outcomes of frail patients [39,40,41,42]. In this context, technological advances and digital solutions [10,43] might have a role in the tailored management of PD patients although several barriers still affect its implementation in routine clinical settings [10,44,45,46,47].

As a result, specific rehabilitation protocols are mandatory not only to address motor symptoms [48] but also to contribute substantially to enhancing stability functioning in individuals with PD and reducing the risk of falls. On the other hand, our findings underlined a significant change in 6MWT after rehabilitation in DBS group (Median (IQR) T1–T0: 81 m (37.3 to 132.3); *p* < 0.0001), suggesting improved functional exercise capacity and endurance. These data emphasized the program’s efficacy in physical performance, a crucial aspect of overall well-being and ADLs for individuals with PD [49]. Exploring the impact on independence in ADLs, the SPDDS captures self-reported abilities to perform ADLs, offering a patient-centric perspective [30]. Interestingly, our data underlined that the rehabilitation program positively affects both study group’s SPDDS scores, suggesting significant implications for the participants’ independence in managing daily tasks. This is a crucial aspect of PD management, as maintaining autonomy in ADLs significantly contributes to an improved quality of life, but also to reduced assistance costs [50].

These positive findings might be partly related to the widely noted effects of rehabilitation in neurodegenerative disorders, leveraging the brain’s inherent capacity for neuroplasticity [51,52,53]. In patients with PD, tailored rehabilitation programs not only target motor symptoms but also engage neural networks, fostering neuroplastic changes that enhance functional outcomes [54].

Recently, growing attention has focused on the integration of technological solutions in rehabilitation, with recent reports suggesting the potential role of transcranial stimulation methods (including transcranial magnetic stimulation (TMS) and transcranial direct current stimulation (tDCS)) in modulating neural activity and enhancing neuroplasticity [55,56]. On the other hand, a recent hypothesis suggested that DBS, while primarily known for alleviating motor symptoms through targeted electrical stimulation, may contribute to neuroplasticity by modulating neural circuits [57,58]. The interaction between rehabilitation and DBS holds promise in augmenting the brain’s adaptive mechanisms, potentially enhancing the effectiveness of rehabilitation interventions [21]. On the other hand, it should be noted that transcranial stimulation techniques often target cortical areas, whereas DBS modulates subcortical structures. Despite these differences, evidence suggests that both approaches may engage overlapping neural networks and trigger neuroplastic responses [57,58]. Understanding these complex pathways is crucial for optimizing treatment strategies and offering a holistic approach to neurorehabilitation in neurological disorders. Further research into the synergistic effects of rehabilitation and DBS on neuroplasticity is necessary for improving patient outcomes and optimizing comprehensive therapeutic intervention.

Altogether, our findings provided positive insight into the effects of a specific rehabilitation intervention in patients with PD after years from DBS implant. The comparative analysis between the DBS Group and the no-DBS Group revealed no significant differences in the variation of outcomes, emphasizing its potential as a standalone intervention or as a complementary approach to DBS. To the best of our knowledge, this is the first study that assessed the role of rehabilitation intervention years after DBS implant. Thus, our findings might contribute to the limited literature on the specific effects of rehabilitation in PD patients post-DBS. The comprehensive assessment of physical function, motor performance, and ADLs provides a comprehensive understanding of the intervention’s impact.

Despite these positive considerations, certain limitations should be acknowledged. The intrinsic limitation of the retrospective design might affect the implications of the study results. In addition, considering the significant evidence for sexual dimorphism in Parkinson’s disease (PD), it would be interesting to investigate potential differences in the effects of rehabilitation between male and female patients with Deep Brain Stimulation (DBS) versus no-DBS groups. Unfortunately, due to the small sample including only nine female patients in both groups, subgroup analysis based on gender was not possible. Lastly, the absence of a control group without rehabilitation limits the ability to isolate the intervention’s effects. However, it should be noted that this manuscript presents the first data available on the long-term effects of DBS on rehabilitation response. Future prospective studies with larger sample sizes and longer follow-up periods can further elucidate the interactions between DBS and rehabilitation.

## 5. Conclusions

Altogether, our findings suggest that rehabilitation is a safe and effective intervention for patients with PD years after DBS implantation. Notably, PD patients with and without DBS showed similar improvement in terms of physical functioning, physical performance, and independence in ADLs, suggesting that long-term DBS effects did not affect rehabilitation outcomes. While the beneficial impact of rehabilitation of patients with PD is widely acknowledged, our study contributes to supporting its role in every phase of PD independently by DBS implantation. Nevertheless, further prospective studies are necessary to confirm our data in order to guide clinicians in optimizing rehabilitation programs and improving overall rehabilitation outcomes in PD patients with DBS.

## Figures and Tables

**Figure 1 medicina-60-00927-f001:**
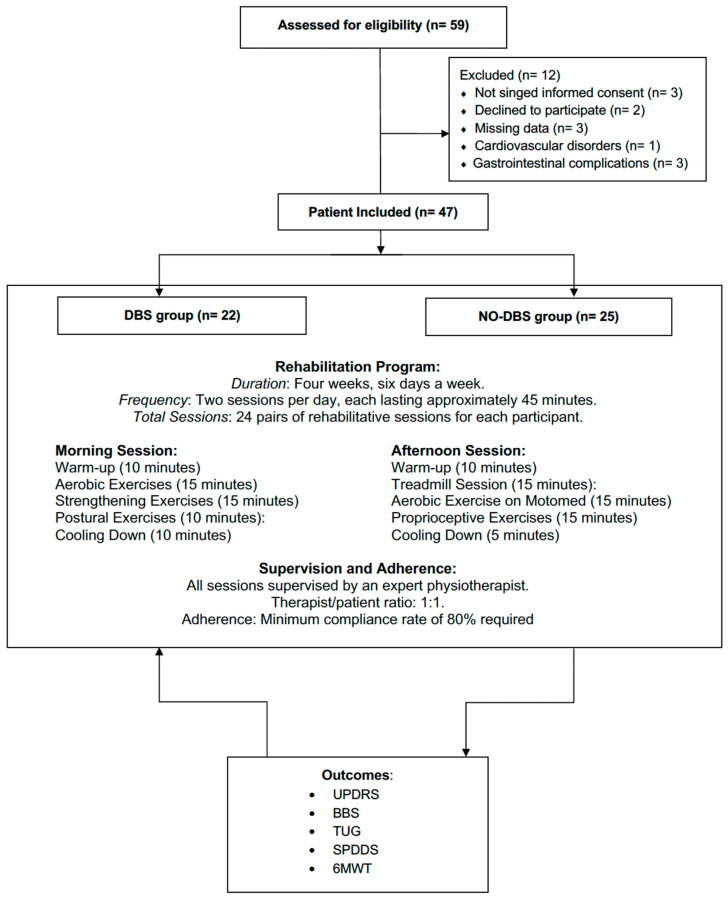
Study flow chart.

**Figure 2 medicina-60-00927-f002:**
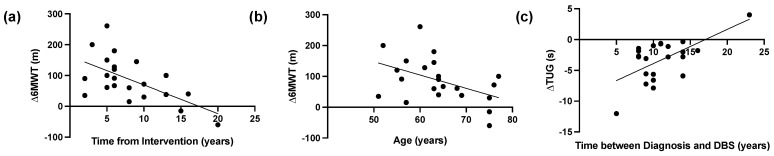
Correlation analysis in the Deep Brain Stimulation (DBS) Group. (**a**) Linear correlation between time from DBS intervention and 6-min walk test (6MWT) improvements (r = −0.516, *p* = 0.014), and (**b**) linear correlation between age at the hospitalization and 6-min walk test (6MWT) improvements (r = −0.427, *p* = 0.047), and (**c**) Linear correlation between years between diagnosis and DBS implant and Timed Up and Go (TUG) (r = 0.434, *p* = 0.044).

**Table 1 medicina-60-00927-t001:** Study population characteristics.

	Whole Sample (*n* = 47)		Groups				
			DBS Group (*n* = 22)		No-DBS Group (*n* = 25)		Between-Group Analysis (*p*-Value)
Age	65.0	61.0 to 74.0	63.5	57.0 to 70.5	69.0	63.0 to 74.0	0.124
Sex (female/male)							0.321
* Female*	14	(29.8%)	9	(22.7%)	9	(36%)	
* Male*	33	(70.2%)	16	(77.3%)	16	(64%)	
Caucasian ethnicity	65	(100%)	22	(100%)	22	(100%)	0.999
Levodopa (mg/day)	500	300 to 800	500	200 to 787.5	500	375.0 to 825.0	0.578
Disease Duration (years)	16.0	13.0 to 19.0	17.0	13.75 to 22.75	15.0	12.0 to 18.0	0.084
Hoehn and Yahr							
* Stage 2*	19	(40.4%)	7	(31.8%)	12	(48%)	0.373
* Stage 3*	28	(59.6%)	15	(68.2%)	13	(52%)	
Age at the DBS implant	-	-	56.0	50.8 to 61.5	-	-	
Years between diagnosis and DBS	-	-	10.0	8.8 to 14.0	-	-	
Years from the intervention	-	-	6.0	5.0 to 10.8	-	-	
MDS-UPDRS	89.0	68.0 to 107.0	105.5	68.75 to 114.3	86.0	66.0 to 96.0	0.072
BBS	45.0	31.0 to 51.0	41.0	29.75 to 51.25	45.0	35.5 to 50.5	0.337
TUG (s)	10.9	8.69 to 18.2	12.7	8.9 to 21.1	9.7	8.3 to 13.1	0.431
SPDDS	57.0	45.0 to 72.0	63.0	47.3 to 75.0	51.0	40.5 to 67.5	0.400
6MWT (m)	310.0	240.0 to 420.0	251.5	175.0 to 427.5	340.0	300.0 to 422.5	0.193
MoCA	22.52	19.98 to 25.19	21.05	18.43 to 32.08	23.52	21.11 to 25.52	0.078

Continuous variables are presented as median with interquartile range, categorical variables are expressed as counts (percentages). *Abbreviations:* 6MWT: 6 M Walking Test; BBS: Berg Balance Scale; MDS-UPDRS: Movement Disorder Society—Unified Parkinson’s Disease Rating Scale; MoCA: Montreal Cognitive Assessment; SPDDS: Self-Assessment Parkinson’s Disease Disability Scale; TUG: Time Up and Go.

**Table 2 medicina-60-00927-t002:** Within-group and between-group differences in outcome measures.

Outcome	DBS Group (*n* = 22)		No-DBS Group (*n* = 25)		
	T1–T0	Within-Group Differences	T1-T0	Within-Group Differences	Between-Group Differences
	Median (IQR)	*p* Value	Median (IQR)	*p* Value	*p* Value
MDS-UPDRS overall score	−16.0 (−32.75 to −7.0)	0.001	−17.0 (−23.0 to −11.5)	<0.0001	0.898
MDS-UPDRS III	−7.0 (−11.5 to −1.0)	0.001	−7.0 (−12.0 to −2.0)	<0.0001	0.856
MDS-UPDRS I II IV	−12.0 (−19.0 to −4.5)	0.001	−10.0 (−14.5 to −5.0)	<0.0001	0.501
BBS	7.0 (3.8 to 10.3)	<0.0001	6.0 (3.5 to 9.5)	<0.0001	0.578
TUG (s)	−2.8 (−5.7 to −1.1)	<0.0001	−2.1 (−3.4 to −1.2)	<0.0001	0.693
SPDDS	−8 (−13.0 to −4.0)	<0.0001	−6.0 (−8.0 to −3.0)	<0.0001	0.174
6MWT (m)	81 (37.3 to 132.3)	<0.0001	83.0 (39.0 to 121.0)	<0.0001	0.983

*Abbreviations*: MDS-UPDRS: Movement Disorder Society—Unified Parkinson’s Disease Rating Scale; BBS: Berg Balance Scale; TUG: Time Up and Go; SPDDS: Self-Assessment Parkinson’s Disease Disability Scale; 6MWT: 6 Meters Walking Test; T0: Baseline; T1: After rehabilitation.

## Data Availability

Data are available on request.

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
