# Peer review of "Long-Term Impact of Deep Brain Stimulation in Parkinson’s Disease: Does It Affect Rehabilitation Outcomes?"

_medicina, 2024, doi:10.3390/medicina60060927_

Round 1
Reviewer 1 Report
Comments and Suggestions for Authors
This study was investigated that the long-term effects of rehabilitation in PD patients underwent DBS. As a results, DBS group was significantly improved. This paper raises several concerns. In particular, it is about long-term effects. Please clarify the definition of long-term effects.
1. P1, line 40-41, The author will need to be specific about motor and non-motor symptoms.
2. P2, line 55, The author should add more details about the mechanism and effects of DBS.
3. P4, line 175, In recent years, MDS-UPDRS, rather than UPDRS, has become the norm. Why did you use UPDRS? However, the citation is MDS. Clearly.
4. P5, line 218, This study was evaluated 24 hours after rehabilitation. Could you please define long term effect, after 24 hours would not be short term?
5. P7, Table 1, It is important to indicate the L-dopa dose and duration of disease in order to indicate the effect of rehabilitation. However, they are not listed in Table 1.
6. P8, line 273, Walking, balance and 6MD are strongly influenced by age. A partial correlation analysis with age as a control factor should be considered.
7. What does UPDRS I II IV indicate? This should be listed separately. Additionally, the overall score should be included.
Comments on the Quality of English Language
No comments.
Author Response
Dear Reviewer,
Many thanks for your letter and kind comments concerning our manuscript. In this manuscript, we provide data about patients at years from DBS implant. The long-term effects of DBS implants represent a critical issue in current literature highlighting cognitive impairment after years from DBS implant, suggesting negative long-term effects of this intervention. However, no previous study characterized how patients with years of DBS stimulation might have benefits from rehabilitation. Thus, we aimed to characterize long-term effects of DBS in terms of rehabilitation response. We acknowledge that both the Title and the Manuscript might be somehow misleading, potentially causing readers to misunderstand “rehabilitation long-term effects” rather than “DBS long-term effects on rehabilitation.” We improved both the title and the manuscript in better characterizing the definition of long-term effects in accordance with the Reviewer’s comment.
- P1, line 40-41, The author will need to be specific about motor and non-motor symptoms.
We would like to thank the Reviewer for the insightful comment. We better characterized about motor and non-motor symptoms in accordance with the Reviewer’s comment.
- P2, line 55, The author should add more details about the mechanism and effects of DBS.
We would like to thank the Reviewer for the insightful comment. We improved the introduction section providing more insights about the mechanism and effects of DBS, in accordance with the Reviewer’s comment.
- P4, line 175, In recent years, MDS-UPDRS, rather than UPDRS, has become the norm. Why did you use UPDRS? However, the citation is MDS. Clearly.
We would like to thank the Reviewer for the insightful comment. We used the MDS-UPDRS in accordance with the current clinical practice and with the reference. We corrected the typo across the manuscript in accordance with the Reviewer’s comment.
- P5, line 218, This study was evaluated 24 hours after rehabilitation. Could you please define long term effect, after 24 hours would not be short term?
We would like to thank the Reviewer for the insightful comment. The long-term effects of DBS implants represent a critical issue in current literature highlighting cognitive impairment after years from DBS implant, suggesting negative long-term effects of this intervention. However, no previous study characterized how patients with years of DBS stimulation might have benefits from rehabilitation. Thus, we aimed to characterize long-term effects of DBS in terms of rehabilitation response. We acknowledge that both the Title and the Manuscript might be somehow misleading, potentially causing readers to misunderstand “rehabilitation long-term effects” rather than “DBS long-term effects on rehabilitation.” We improved both the title and the manuscript in better characterizing the definition of long-term effects in accordance with the Reviewer’s comment.
- P7, Table 1, It is important to indicate the L-dopa dose and duration of disease in order to indicate the effect of rehabilitation. However, they are not listed in Table 1.
We would like to thank the Reviewer for the insightful comment. We included data about the L-dopa dose and duration of disease to better clarify the sample characteristics in accordance with the Reviewers’ comment.
- P8, line 273, Walking, balance and 6MD are strongly influenced by age. A partial correlation analysis with age as a control factor should be considered.
We would like to thank the Reviewer for the insightful comment. A partial correlation analysis with age has been performed and included in Supplementary Table 1. We also improved the results section in accordance with the Reviewer’s comment.
- What does UPDRS I II IV indicate? This should be listed separately. Additionally, the overall score should be included.
We would like to thank the Reviewer for the insightful comment. We better characterized that MDS-UPDRS I II IV provided data about Parkinson’s’ Disease non-motor symptoms. Moreover, we included in the Table the MDS-UPDRS overall score.
Reviewer 2 Report
Comments and Suggestions for Authors
Summary: This single-center, retrospective study assessed the impact of long-term rehabilitation intervention in patients with Parkinson’s Disease (PD) across groups treated and not treated with deep brain stimulation (DBS). Patients were assessed using different tools for PD progression, static and dynamic balance, functional mobility, exercising, and cognitive ability. The study shows that rehabilitation led to improvements in all assessments in DBS and non-DBS groups. The authors describe that physical rehabilitation led to similar improvements from baseline in both DBS vs non-DBS groups. The study adds to an existing but limited body of literature showing that physical rehabilitation of varying durations tends to improve functional outcomes for patients with PD (PMID: 19398215, 35992727), with the addition of multiple measures to test the improvement.
However, there are some major revisions required:
Major comments: Between-group comparison: The study compares the DBS and non-DBS groups in terms of improvement in physical function after rehabilitation treatment, which shows similar outcomes. However, in order to provide the reader with a more comprehensive comparison, the authors should consider performing an analysis between the baseline assessment of the DBS and non-DBS groups. This would help in assessing if DBS treatment had any effect on patients’ function before physical rehabilitation. Additionally, it seems that the authors want to conclude that rehabilitation has the potential to be a standalone intervention or a complementary approach to DBS. However, presenting baseline data is imperative to understanding the role of DBS in the studied functional endpoints.
Male vs Female patients: Since there has been strong evidence for sexual dimorphism in PD, it would be helpful if the authors could describe if the effects of rehabilitation in the DBS and non-DBS groups differ between male and female patients.
Minot Comments:
There seems to be a difference in the labeling of groups in Table 1 vs Table 2 and the text. In Table 1, the comparator group is labeled control, whereas elsewhere it is labeled non-DBS, similar terminologies should be used across the manuscript.
Line 316: There seems to be an error in the p-value reported for the 6MWT, please clarify.
Line 50-51: Authors cite ref 9 describing rehabilitation playing a pivotal role in managing symptoms in patients with PD. However, it seems that ref 9 describes the effect of DBS on learning in older adults with or without PD. Please check.
Line 251-252: There seems to be a mismatch in the reported median age of the non-DBS group in the text vs. that in the table. Please clarify.
Line 263-269: The paragraph mentions an improvement in all assessments for the DBS group after rehabilitation, that is the within-group difference from baseline to after 4 weeks of rehabilitation. However, a similar improvement is observed for the non-DBS group as well. The authors should consider including a statement explaining the data for the non-DBS group in the text.
Table 2: There seems to be an error in the within-group p-values reported for both the comparator groups, please clarify what does p values= 0.000 signify.
Line 315-316: It seems that the authors claim that the median 6mWT distance is 81m, however, it is unclear which patient group the data refers to. Additionally, as per Table 2, 81m is the change in 6MWT post-rehabilitation and not an absolute data point. Please check and clarify.
Comments on the Quality of English Language
Overall English language seems fine. There are instances where the the sentences is confusing which needs to be improved.
Author Response
Dear Reviewer
Many thanks for your letter and kind comments concerning our manuscript. We have revised our manuscript carefully based on suggestions and answered the Reviewers highlighting the corrections in the text.
Between-group comparison: The study compares the DBS and non-DBS groups in terms of improvement in physical function after rehabilitation treatment, which shows similar outcomes. However, in order to provide the reader with a more comprehensive comparison, the authors should consider performing an analysis between the baseline assessment of the DBS and non-DBS groups. This would help in assessing if DBS treatment had any effect on patients’ function before physical rehabilitation. Additionally, it seems that the authors want to conclude that rehabilitation has the potential to be a standalone intervention or a complementary approach to DBS. However, presenting baseline data is imperative to understanding the role of DBS in the studied functional endpoints.
We would like to thank the Reviewer for the insightful comment. We better characterized the between-groups analysis of the baseline outcome measure in the DBS and non-DBS groups within Table 1, in accordance with the Reviewer’s suggestion.
Male vs Female patients: Since there has been strong evidence for sexual dimorphism in PD, it would be helpful if the authors could describe if the effects of rehabilitation in the DBS and non-DBS groups differ between male and female patients.
We would like to thank the Reviewer for the insightful comment. Considering the significant evidence for sexual dimorphism in Parkinson's disease (PD), we acknowledge the importance of investigating potential differences in the effects of rehabilitation between male and female patients, particularly in the context of Deep Brain Stimulation (DBS) versus non-DBS groups. Unfortunately, due to the small sample including only 9 female patients in both groups, subgroup analysis based on gender was not possible. However, it should be noted that this manuscript presents the first data available on the long-term effects of DBS on rehabilitation outcomes. Further studies might characterize the role of gender on rehabilitation response. We improved the limitation section highlighting the critical issue that the Reviewer pointed out in order to improve the paper in accordance with the Reviewer’s comment.
There seems to be a difference in the labeling of groups in Table 1 vs Table 2 and the text. In Table 1, the comparator group is labeled control, whereas elsewhere it is labeled non-DBS, similar terminologies should be used across the manuscript.
We would like to thank the Reviewer for the insightful comment. We improved the tables in accordance with the Reviewer’s comment.
Line 316: There seems to be an error in the p-value reported for the 6MWT, please clarify.
We would like to thank the Reviewer for the insightful comment. We improved both Table and manuscript better characterizing the p-value p<0.0001 rather than p=0.000 in accordance with the Reviewer’s instructions.
Line 50-51: Authors cite ref 9 describing rehabilitation playing a pivotal role in managing symptoms in patients with PD. However, it seems that ref 9 describes the effect of DBS on learning in older adults with or without PD. Please check
We would like to thank the Reviewer for the insightful comment We have replaced the reference with a more suitable one (Abbruzzese G, Marchese R, Avanzino L, Pelosin E. Rehabilitation for Parkinson's disease: Current outlook and future challenges. Parkinsonism Relat Disord. 2016 Jan;22 Suppl 1:S60-4. doi: 10.1016/j.parkreldis.2015.09.005. Epub 2015 Sep 3. PMID: 26360239.”).
Line 251-252: There seems to be a mismatch in the reported median age of the non-DBS group in the text vs. that in the table. Please clarify.
We would like to thank the Reviewer for the insightful comment. We corrected the typo in the text accordingly.
Line 263-269: The paragraph mentions an improvement in all assessments for the DBS group after rehabilitation, that is the within-group difference from baseline to after 4 weeks of rehabilitation. However, a similar improvement is observed for the non-DBS group as well. The authors should consider including a statement explaining the data for the non-DBS group in the text.
We would like to thank the Reviewer for the insightful comment. We improved the data presentation clarifying that both groups had significant improvements in the outcome measures, without significant differences between groups.
Table 2: There seems to be an error in the within-group p-values reported for both the comparator groups, please clarify what does p values= 0.000 signify.
We would like to thank the Reviewer for the insightful comment. We improved both Table and manuscript better characterizing the p-value p<0.0001 rather than p=0.000 in accordance with the Reviewer’s instructions.
Line 315-316: It seems that the authors claim that the median 6mWT distance is 81m, however, it is unclear which patient group the data refers to. Additionally, as per Table 2, 81m is the change in 6MWT post-rehabilitation and not an absolute data point. Please check and clarify.
We would like to thank the Reviewer for the insightful comment. We improved the Discussion Section characterizing that 81m is the change in 6MWT post-rehabilitation in DBS group, in accordance with the reviewer’s comment.
Round 2
Reviewer 2 Report
Comments and Suggestions for Authors
Thanks for addressing all the comments. Congratulations! and all the best for future endeavors.